# Detecting temporal and spatial malaria patterns from first antenatal care visits

Arnau Pujol [1,10], Nanna Brokhattingen [1,10], Glória Matambisso[2,10], Henriques Mbeve[2], Pau Cisteró [1], Anna Escoda[1], Sónia Maculuve[2], Boaventura Cuna[2], Cardoso Melembe[2], Nelo Ndimande[2], Humberto Munguambe[2], Júlia Montaña [1], Lídia Nhamússua [2], Wilson Simone [2], Kevin K. A. Tetteh [3], Chris Drakeley [3], Benoit Gamain [4], Chetan E. Chitnis[5], Virander Chauhan [6], Llorenç Quintó[1,2], Arlindo Chidimatembue[2], Helena Martí-Soler[1], Beatriz Galatas [1,2], Caterina Guinovart [1], Francisco Saúte[2], Pedro Aide [2,7], Eusébio Macete[2] & Alfredo Mayor [1,2,8,9] ✉

Pregnant women attending first antenatal care (ANC) visits represent a promising malaria surveillance target in Sub-Saharan Africa. We assessed the spatio-temporal relationship between malaria trends at ANC ($n = 6471$) and in children in the community ($n = 3933$) and at health facilities ($n = 15,467$) in southern Mozambique (2016–2019). ANC *P. falciparum* rates detected by quantitative polymerase chain reaction mirrored rates in children, regardless of gravidity and HIV status (Pearson correlation coefficient [PCC] > 0.8, $\chi^2 < 1.1$), with a 2–3 months lag. Only at rapid diagnostic test detection limits at moderate-to-high transmission, did multigravidae show lower rates than children (PCC = 0.61, 95%CI[−0.12−0.94]). Seroprevalence against the pregnancy-specific antigen VAR2CSA reflected declining malaria trends (PCC = 0.74, 95%CI[0.24−0.77]). 60% (9/15) of hotspots detected from health facility data ($n = 6662$) using a novel hotspot detector, EpiFRIenDs, were also identified with ANC data ($n = 3616$). Taken together, we show that ANC-based malaria surveillance offers contemporary information on temporal trends and geographic distribution of malaria burden in the community.

Surveillance is key to inform optimal and equitable resource allocation for malaria control and elimination[1]. Estimating malaria trends from clinical cases at health facilities remains challenging due to differences in care-seeking behaviour, unknown denominator populations, and asymptomatic infections[2]. These biases are minimised in nationally-representative cross-sectional surveys, but due to their high costs and complex logistics, they are typically only conducted every 2–3 years[3].

Pregnant women attending a first antenatal care (ANC) visit have been proposed as a potential convenience group for surveillance of malaria and other infectious diseases[3–5]. In sub-Saharan Africa, 79% of

[1]ISGlobal, Hospital Clínic, Universitat de Barcelona, Barcelona, Spain. [2]Centro de Investigação em Saúde de Manhiça (CISM), Maputo, Mozambique. [3]Faculty of Infectious and Tropical Diseases, London School of Hygiene & Tropical Medicine, London, UK. [4]Université Paris Cité, INSERM, BIGR, F-75014 Paris, France. [5]Malaria Parasite Biology and Vaccines, Department of Parasites & Insect Vectors, Institut Pasteur, Paris, France. [6]Malaria Group, International Centre for Genetic Engineering and Biotechnology (ICGEB), New Delhi, India. [7]National Institute of Health, Ministry of Health, Maputo, Mozambique. [8]Spanish Consortium for Research in Epidemiology and Public Health (CIBERESP), Madrid, Spain. [9]Department of Physiologic Sciences, Faculty of Medicine, Universidade Eduardo Mondlane, Maputo, Mozambique. [10]These authors contributed equally: Arnau Pujol, Nanna Brokhattingen, Glória Matambisso. ✉e-mail: alfredo.mayor@isglobal.org

pregnant women attend at least one ANC visit[6], offering a good representation of the population. Since visits are unrelated to illness, malaria testing is not biased by care-seeking behaviour or testing decisions, and captures asymptomatic infections[7].

A meta-analysis of pooled prevalence data from Sub-Saharan Africa found a strong correlation between malaria burden in pregnant women and children, with lower rates in the later but with large heterogeneity between studies[8]. Less heterogeneity was found for low-prevalence settings (prevalence <5%), and less difference between women and children was found when restricting the analysis to primigravidae. However, only one study recruited women from an antenatal clinic, and small-scale trends could not be assessed due to pooling data obtained from different administrative levels. Studies using routine ANC data in Tanzania did not provide information on gravidity and were unable to reproduce a similar linear effect as the meta-analysis when comparing young women to children[9–11]. Another study analysed data from health centres in conflict settings in the Democratic Republic of Congo and found a strong but non-linear relationship between ANC prevalence and incidence in children[12]. However, all the studies used low-sensitivity microscopy or rapid diagnostic tests (RDT), missing the significant proportion of *P. falciparum* infections with parasite densities below the detection threshold of conventional field diagnostic tools[13]. Moreover, the effect of gravidity and other factors such as HIV was not consistently assessed. Also, spatial patterns in malaria burden have not been compared between both groups due to lack of geospatial data. Finally, none of the studies quantified correlation in low-transmission settings, where ANC-based surveillance might be particularly attractive due to local clustering of malaria cases which are difficult to monitor with traditional strategies[2]. Therefore, a better understanding of the validity of ANC prevalence data for monitoring transmission in the community and the factors that affect this relationship remains to be developed.

New surveillance tools, such as antibodies against the pregnancy-specific antigen VAR2CSA that mediates parasite sequestration in the placenta[14–16], can potentially increase sensitivity to detect recent exposure in low-transmission settings where detecting a sufficient number of active infections to estimate trends in burden is difficult[16,17]. Combined with novel clustering approaches, ANC data can increase the resolution to detect spatial patterns[3], providing a cost-effective approach to target interventions to the most affected areas.

In this study, we estimate and compare malaria burden at first ANC visits with data from cross-sectional surveys and clinical cases in three settings in southern Mozambique with different transmission levels. We correlate temporal and spatial trends at both RDT- and quantitative polymerase chain reaction (qPCR)-detection levels, and characterise the effect of HIV and gravidity on the correlation. Finally, we assess the added value of antibody data obtained from a bead-based multiplex immunoassay against VAR2CSA and general malaria antigens, and a newly developed hotspot detection algorithm, as innovative tools to improve surveillance in malaria-endemic areas.

## Results

### *P. falciparum* burden
*P. falciparum* parasite rates using qPCR ($Pf$PR$_{qPCR}$) in pregnant women and in children 2–10 years old were highly correlated (Pearson correlation coefficient [PCC] = 0.94 [95% CI 0.70–0.99]) and showed a consistent and linear relationship (slope = 0.97 [95% CI 0.53–2.14], origin = 0.03 [95% CI −0.01–0.08]). Similar high correlations (PCC > 0.85), linear relationships (slope-1 and origin-0) and consistencies ($\chi^2$<1.1) were found regardless of gravidity and HIV status (Fig. 1a–c, Supplementary Fig 1a–c and Supplementary Table 1). At RDT-detection levels, *P. falciparum* parasite rates ($Pf$PR$_{RDT}$) in pregnant women and children showed lower correlations and weaker linear relationships (Fig. 1d–f, Supplementary Fig. 1d–f and Supplementary Table 1). Only primigravidae showed a 1-to-1 linear relationship with children (Fig. 1e),

whereas multigravidae showed lower rates that were not correlated (PCC = 0.61 [95% CI −0.12–0.94]), with a linear regression slope not consistent with equality (0.17 [95% CI −0.035–0.49]; Fig. 1f). However, in low-transmission Magude and Manhiça, good consistency of both $Pf$PR$_{qPCR}$ and $Pf$PR$_{RDT}$ between multigravid pregnant women and children was observed ($\chi^2$ < 1.10), regardless of HIV status (Fig. 1g–i, Supplementary Fig. 1g–i).

### *P. falciparum* temporal trends
Overall, $Pf$PR$_{qPCR}$ in pregnant women declined from 10.7% to 4.1% during the study, resembling an overall decline of 62% (65%, 58%, and 62% in Magude, Ilha Josina, Manhiça, respectively, $p < 0.001$; Fig. 2a and Supplementary Table 2). A similar decline of 60% (from 4.4% to 1.8%) was observed for $Pf$PR$_{RDT}$ (60% in both Ilha Josina and Manhiça, $p = 0.001$; 52% in Magude, $p = 0.034$). Clinical cases showed an overall decline of 47% (73%, 38 and 52% in Magude, Ilha Josina and Manhiça, respectively, $p < 0.001$). Statistically significant declines in parasite rates among children 2–10 years old from cross-sectional surveys were only observed in Ilha Josina for $Pf$PR$_{qPCR}$ (90%, $p = 0.004$). The $Pf$PR$_{qPCR}$ temporal patterns in pregnant women and clinical cases were consistent ($\chi^2 < 1$) and correlated (PCC = 0.87 [95%CI 0.69–0.91]) with a delay of ~90 days, regardless of gravidity (Fig. 2b, d, f and Table 1), meaning that the best correlation was obtained between clinical cases and ANC data collected ~90 days after the clinical cases. Similar results were found at RDT-detection levels (PCC > 0.78 [95%CI 0.27–0.85]; Fig. 2c, e, g), but with shorter time lags (~40 days) when including multigravida women. HIV did not significantly impact estimates (Supplementary Fig. 2).

### Spatial trends
Pregnant women and children 2–10 years old showed consistent 2-point correlation function (2PCF) statistics ($\chi^2$<1.50) in all three years, regardless of gravidity and HIV status at both qPCR and RDT-detection levels (Fig. 3a–c, Supplementary Fig 3 and Supplementary Table 3). 2PCFs from ANC data significantly deviated from 0 (Supplementary Table 4), being negative at distances between sample pairs within and between Magude and Manhica (<10 km and ~45 km, respectively), and positive at distances between Ilha Josina and north of Manhiça (~30 km).

### *P. falciparum* hotspots
The performance of Epidemiological Foci Relating Infections by Distance (EpiFRIenDs), a novel malaria hotspot detector, and SaTScan to detect clusters on simulated data was compared for three different scenarios (Supplementary Methods: EpiFRIenDs). In the first scenario with a random spatial distribution of positive and negative cases, the only difference between the methods was observed on the detection of false small clusters in small-scale parameterisations of EpiFRIenDs, which can be corrected after a parameter calibration ("Methods" section, Supplementary Methods). In the second scenario, positive cases were correctly identified as part of the four circular clusters in both methods. And in the third scenario, EpiFriends could correctly assign all positive cases as part of the sinusoidal cluster which could not be detected with SaTScan.

Due to its higher capacity to detect clusters of arbitrary shape, further analysis was conducted using EpiFRIenDs. With this spatial algorithm, 11 hotspots, all in Ilha Josina, were detected with qPCR results from geolocalized pregnant women in Manhiça district ($n = 3616$; Fig. 3d–f). Four of them, and the two most persistent ones, occurred during the first year, when transmission was highest (Fig. 3d and Supplementary Movie 1 and 2). Six hotspots persisted for more than 20 days, one of them for more than 90 days (Fig. 3j–k). Fourteen hotspots were found using the paediatric outpatient morbidity surveillance system (OPMSS) clinical data from Manhiça district (using a 18% sub-sampling of $n = 6662$ visits from a total of 37,131,

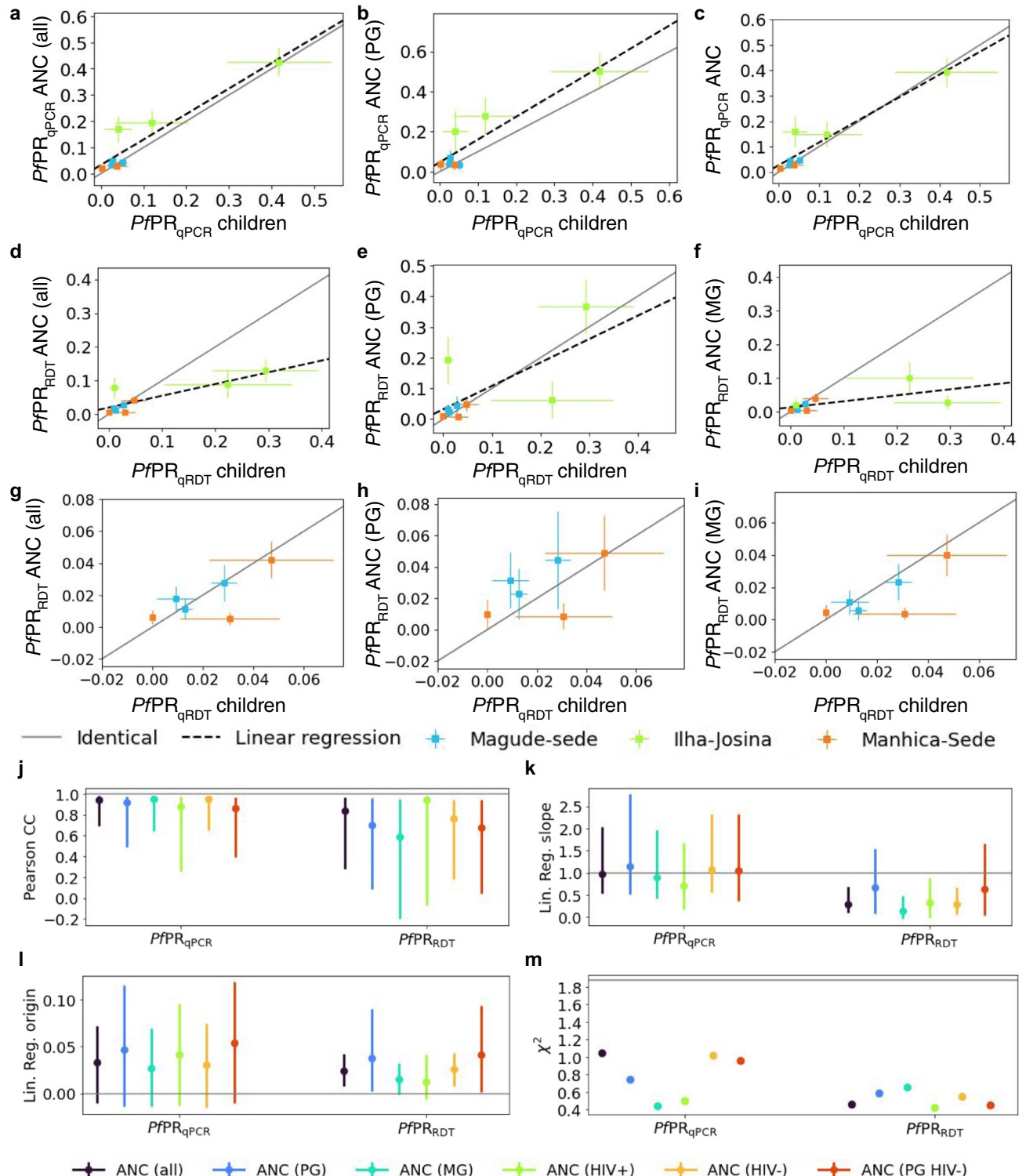

**Fig. 1 | Relationship between *Plasmodium falciparum* parasite rates in pregnant women at first ANC visit and children from cross-sectional surveys.** Scatter comparison of $Pf$PR$_{qPCR}$ (**a**–**c**) and $Pf$PR$_{RDT}$ (**d**–**f**) between children 2–10 years old ($n = 3933$) and **a** women attending a first ANC visit 90 days around the dates of cross-sectional surveys ($n = 2016$), **b** primigravid women (PG) ($n = 602$) and **c** multigravidae (MG) ($n = 1414$). **g**–**i** Same as **d**–**f** but restricted to low transmission areas (Magude and Manhiça) ($n = 3818, 1884, 530, 1354$ for children 2–10 years old, all first ANC visits, PG and MG respectively). The three points with error bars of each colour represent the mean values for each of the three years of study ±SD. Grey lines in **a**–**i** show the hypothetical one-to-one relationship, and the black dashed lines show the linear regressions. **j**–**m** values of linear regression slopes and origin parameters, Pearson correlation coefficients and $\chi^2$ statistics of the relationships between pregnant women and children, with the error bars representing the 95% CI. The horizontal grey lines show the optimal scenario (**j**–**l**) and the threshold of $\chi^2$ value for consistency (**m**).

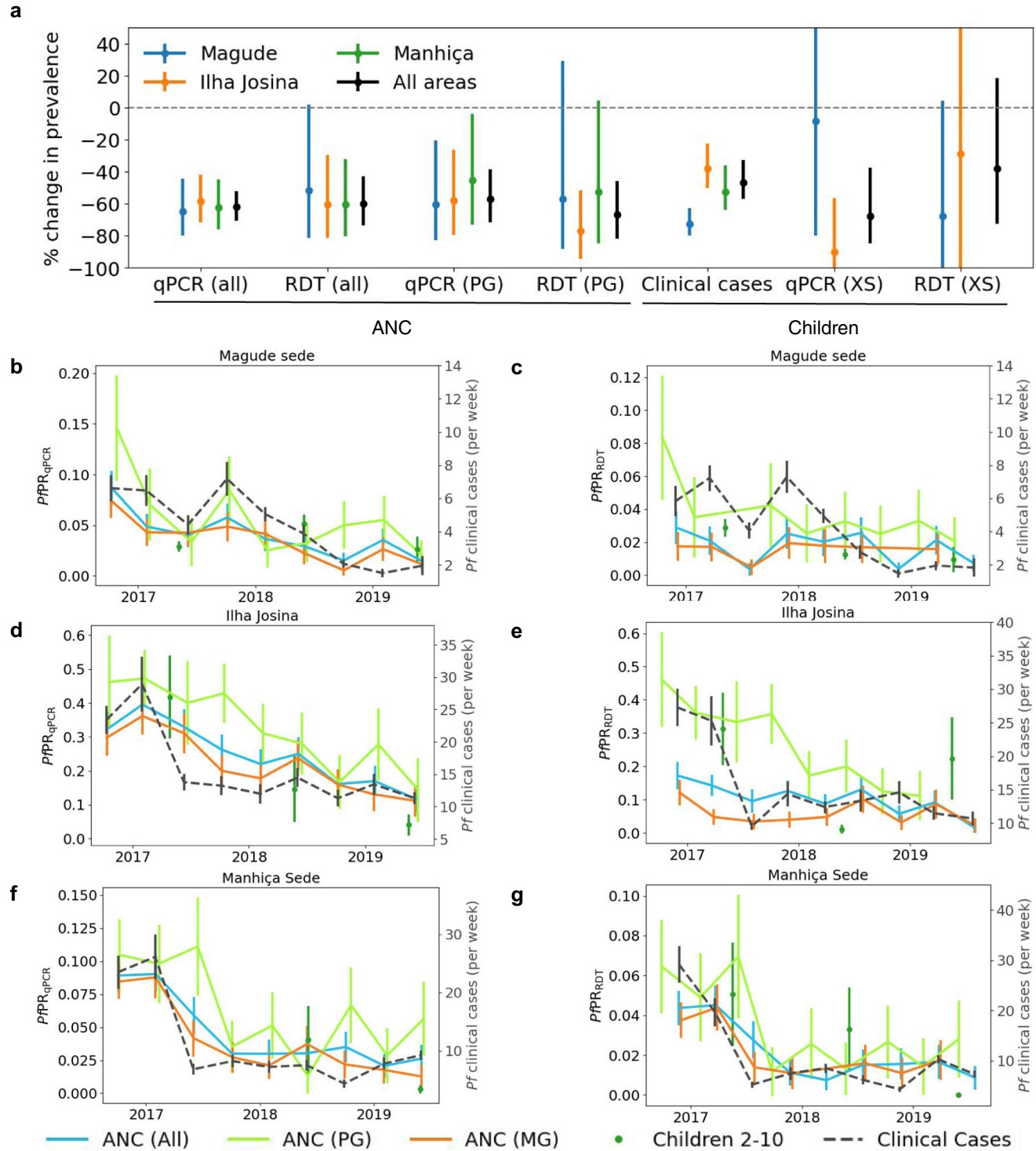

**Fig. 2 | Temporal trends in *Plasmodium falciparum* burden. a** Changes in prevalence of different *Plasmodium falciparum* burden indicators in the three areas (colours) and in total (black) (same sample sizes as for **b**–**g**). Error bars indicate the 95% CI. **b**–**g** Temporal trends in the number of weekly clinical cases (dashed grey lines) (*n* = 15,467 divided in the three areas), positivity rates in children 2–10 years old (*n* = 3933) from cross-sectional surveys (green dots), and in pregnant women at first ANC visit (coloured lines), for the three studied areas (from top to bottom)

(*n* = 6471, 1754 and 4717 for all, PG and MG divided by the three areas). Panels **a**, **c**, **e** show estimates from qPCR results, and panels **b**, **d**, **f** show estimates using the detection limit of RDTs to define parasite rates. Time lags that maximise the Pearson correlation coefficients of the temporal trends between ANC data and clinical cases were applied (see Table 1). PG primigravid women, MG multigravid women, XS cross-sectional surveys. Data are presented as mean values ±SD.

see Supplementary Methods), 11 (79%) of them in Ilha Josina (Fig. 3g–i). Six hotspots persisted for more than 20 days, two of them for more than 40 days. Nine of the hotspots detected from both ANC and clinical data matched spatially (seven with *p* ≤ 0.054, and two with *p* ≤ 0.91;

Supplementary Table 5). Hotspots showed temporal heterogeneity, with 11 (79%) of clinical case hotspots detected during the rainy seasons, while six (55%) ANC-hotspots were detected then. When considering both spatial and temporal proximity between hotspots, seven

**Table 1 | Comparison of *Plasmodium falciparum* temporal patterns estimated from pregnant women at first antenatal care visit and clinical cases in children**

| Test | Population | $\chi^2$ | Pearson CC (95%CI) | Time lag (days) |
|---|---|---|---|---|
| qPCR | | | | |
| | All prenatal | 1 | 0.87 (0.69, 0.91) | 89 |
| | Primigravidae | 0.97 | 0.69 (0.31, 0.81) | 81 |
| | Multigravidae | 0.97 | 0.90 (0.66, 0.92) | 89 |
| | HIV-infected | 0.56 | 0.80 (0.26, 0.86) | 127 |
| | HIV-uninfected | 0.83 | 0.87 (0.64, 0.92) | 89 |
| | Primigravid HIV- | 0.94 | 0.72 (0.24, 0.80) | 37 |
| RDT | | | | |
| | All prenatal | 1.49 | 0.78 (0.27, 0.85) | 39 |
| | Primigravidae | 1.23 | 0.66 (0.14, 0.78) | 91 |
| | Multigravidae | 0.78 | 0.82 (0.24, 0.87) | 33 |
| | HIV-infected | 0.34 | 0.79 (0.06, 0.84) | 43 |
| | HIV-uninfected | 1.14 | 0.71 (0.24, 0.81) | 39 |
| | Primigravid HIV- | 1.06 | 0.66 (0.13, 0.80) | 37 |

Pearson correlation coefficients and $\chi^2$ statistics of the comparison between the temporal trends in $Pf$PR$_{qPCR}$ (first six rows) and $Pf$PR$_{RDT}$ (last six rows) in different populations of pregnant women at first ANC visit and the mean weekly number of clinical cases, with their time lag of optimal correlation. A time lag of 89 days indicates that the results of the clinical cases in children had the best correlation with the results of pregnant women taken 89 days after the clinical cases in children.

hotspots from both ANC and clinical data matched (Supplementary Table 5). With 1-year temporal windows ($n$ = 5049 first ANC visits at the three health facilities), four RDT hotspots (all in Ilha Josina) and 11 qPCR hotspots were found, regardless of whether HIV-positive or multigravid women were included (Supplementary Table 6).

**Seroprevalence of antibodies against *P. falciparum* antigens**
A total of 5,990 dried blood spots (DBS) were succesfully analysed for the presence of antibodies against 14 *P. falciparum* antigens using a quantitative multiplexed bead array (Supplementary Fig 4). The lowest seroprevalence was found for peptide VAR2CSA$_{P8}$ in Magude during the third year (3.9%, 95%CI 3.0–5.1), and the highest seroprevalence was found for MSP1 in Ilha Josina during the first year (76.8%, 95%CI 72.3–80.8) (Supplementary Table 7). ANC-seroprevalences were correlated with $Pf$PR$_{qPCR}$ in children (PCC > 0.7), with antigen-dependent linear regression parameters (Fig. 4a and Supplementary Table 8). The highest correlated antibodies were those against GEXP18, RH2, RH5, VAR2CSA$_{DBL3-4}$, and peptides VAR2CSA PD, P39, and P8 (PCC > 0.85; Fig. 4a, b). Correlations remained high across groups of gravidity and HIV status, however, wide confidence intervals increased to include potentially no correlation for some antibodies (Fig. 4a). Significant declines in seroprevalence across all areas were only observed for VAR2CSA$_{DBL3-4}$ (Fig. 4c and Supplementary Tables 7 and 9). The highest correlation in temporal trends between seroprevalence and clinical cases was found for VAR2CSA$_{DBL3-4}$ (PCC = 0.74 [95%CI 0.24–0.77]) and VAR2CSA$_{P1}$ (PCC = 0.74 [95%CI 0.27–0.76]), remaining high when stratifying by gravidity (Supplementary Table 10). Seroprevalence trends lagged up to 10 months behind clinical cases, with no clear pattern based on pregnancy-specificity or expected longevity of antibodies[18].

2PCF measurements of serostatus agreed well with those from qPCR-detected cases in children from cross-sectional surveys ($\chi^2 \leq 1.6$; Fig. 4e−g and Supplementary Table 11). Similar numbers and locations of seropositivity clusters (seroclusters) and qPCR hotspots were observed for HSP40, Etramp, EBA175, VAR2CSA$_{DBL3-4}$, and combining all VAR2CSA peptides (Supplementary Fig 5d−g and Supplementary Table 6). However, in year 2 in Ilha Josina (after the steep decline in

$Pf$PR$_{qPCR}$), more seroclusters of HSP40, $Pf$Tramp, RH5, VAR2CSA$_{DBL3-4}$ and combined VAR2CSA peptides were found than hotspots. In addition, eight seroclusters were found in year 1 in the south-west of Magude, which did not reflect hotspots of qPCR-positive cases. Finally, in the north of Manhiça we detected one hotspot from qPCR data in year 1. In this same area, two seroclusters (Etramp, VAR2CSA$_{P39}$) were detected in year 1, and 6 others (MSP1, EBA175, ACS5, $Pf$Tramp, VAR2CSA$_{DBL3-4}$ and combining all the VAR2CSA peptides) were detected in year 3. Seroclusters were more stable than qPCR hotspots, with nine persisting more than 20 days and one persisting for 197 days (Supplementary Fig. 6 and Supplementary Movie 3).

## Discussion

This population-based spatio-temporal analysis of parasitological and serological data from southern Mozambique shows that qPCR-positivity rates at first ANC visit reflect rates in children with a time lag of 2–3 months relative to clinical cases, regardless of the women's gravidity, HIV status, and the transmission intensity in their area. Disparities emerge at RDT-detection levels for multigravid women in moderate-to-high transmission settings, indicating the need to consider gravidity in the analysis when using diagnostic tools of limited sensitivity. However, gravidity did not affect the number of hotspots detected which were similar to those detected using passive surveillance data. VAR2CSA seroprevalence at first ANC visit was found to be highly correlated with parasite rates in children, sensitive to temporal and spatial trends that were missed by RDT data in children, and unaffected by gravidity, therefore constituting a robust adjunct for ANC surveillance. Overall, this study demonstrates the value of pregnant women for programmatic surveillance in malaria-endemic regions in sub-Saharan Africa.

Our study provides evidence that parasite rates in pregnant women are highly correlated and consistent with rates in children from cross-sectional surveys at qPCR-detection levels, across the transmission spectrum, and regardless of HIV status and gravidity. The lower RDT-based rates in multigravid women compared with children in high-to-moderate transmission Ilha Josina, in line with previous reports[8,10,11], is probably explained by immunity acquired during successive pregnancies, which enable the control of parasite densities below the RDT-detection limit[19,20]. The bias introduced by infections missed by RDT in multigravid women at moderate-to-high transmission levels can be avoided by restricting the analysis to primigravidae, using highly sensitive detection tools, such as qPCR or VAR2CSA serology, or with models accounting for immunity. Temporal trends in ANC-based parasite rates, both detected by qPCR and RDT, reflected trends in clinical cases observed 2–3 months earlier, similar to the 3-month time lag observed in the Democratic Republic of Congo[12]. This lag suggests that infections at ANC are older than those in symptomatic children, in accordance with previous studies showing that infections in pregnancy mainly result from a boosting of infections acquired before pregnancy[21]. RDT rates in multigravid women showed shorter time lags, possibly due to faster clearance of infections by anti-parasite immunity acquired during previous pregnancies. Although the time lag may limit the use of ANC data to predict changes in clinical cases, it would still be useful to benchmark passive surveillance estimates and population-based cross-sectional surveys, improving population denominators and informing about the burden of asymptomatic infections. On the other hand, the results imply that clinical cases can be used to predict changes in malaria burden among pregnant women at first ANC visit.

Similar spatial patterns of *P. falciparum* cases and hotspots were observed among pregnant women at ANC and children in the community, regardless of the women's gravidity or HIV status. Spatial clustering decreased from year 1 to year 2, reflecting trends in burden. The new software EpiFRIenDs revealed finer spatial structures of *P.*

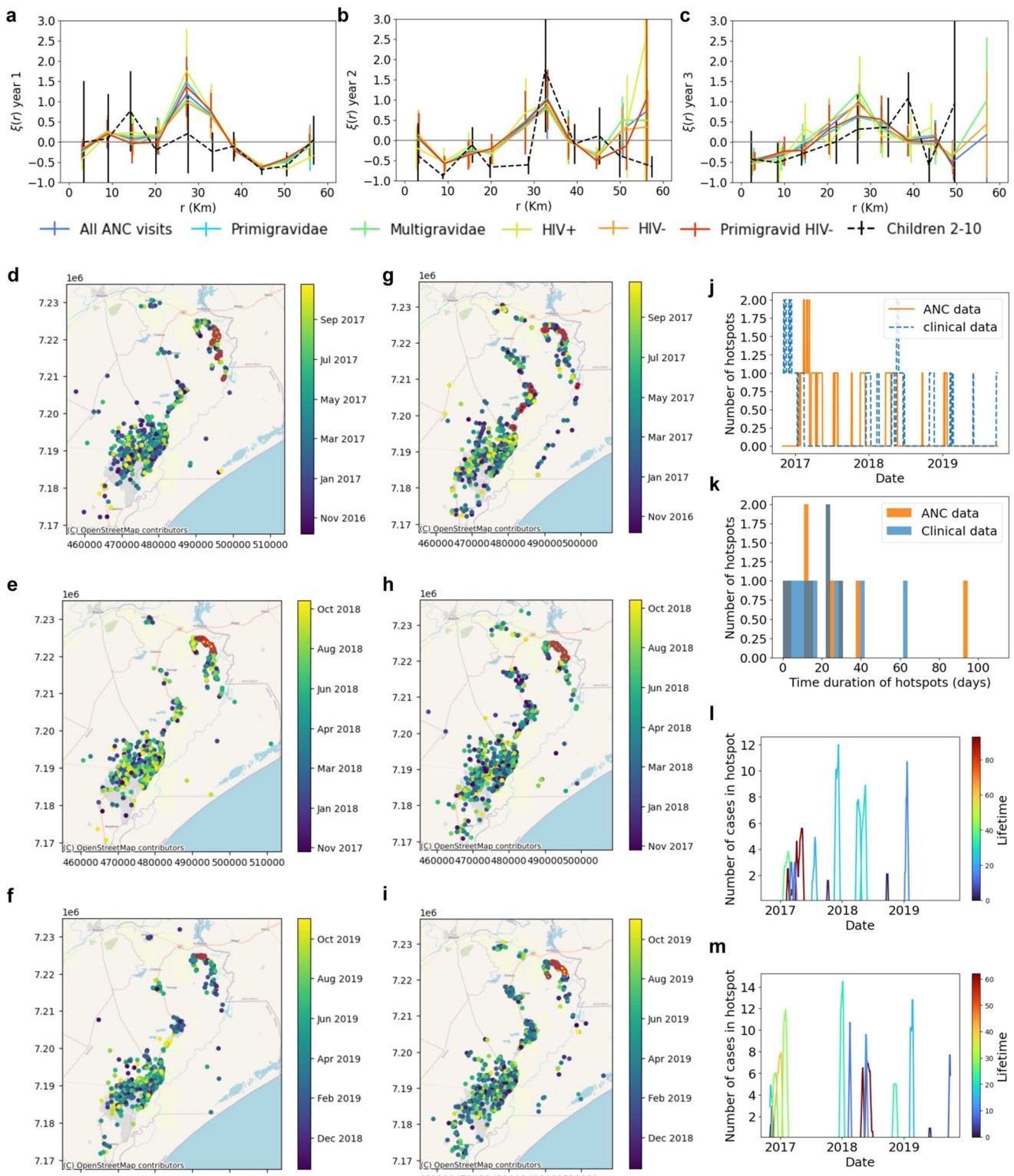

**Fig. 3 | Spatial clustering and hotspots of *Plasmodium falciparum* infections.**
**a**–**c** 2-point correlation functions of *P. falciparum* infections with geospatial data available in pregnant women at first ANC visit (different gravidity and HIV status shown in different colours) (*n* by year= 1971, 1613, 1465) and in children 2–10 years old (*n* by year= 1900, 1565, 468) from cross-sectional surveys (black dashed lines) for the three years and study areas, showing mean values ±SD. **d**–**i** Temporal variation of the spatial distribution of the households from pregnant women at first ANC visits (**d** for year 1, **e** for year 2 and **f** for year 3) and from children attending health facilities (**g** for year 1, **h** for year 2 and **i** for year 3) in Manhiça district (colour coded by their visit date). Cases circled in red belong to hotspots detected using temporal windows of one month. **j** Temporal distribution of the number of hotspots detected from ANC (orange) and from clinical (blue) data. **k** Histogram of lifetimes of identified hotspots from ANC (orange) and from clinical (blue) data. **l**, **m** Timeline of identified hotspots with their size (y-axis) and colour coded by their lifetime from ANC (**l**) and clinical malaria cases (**m**) data. Maps used OpenStreetMap data, available under the Open Database License.

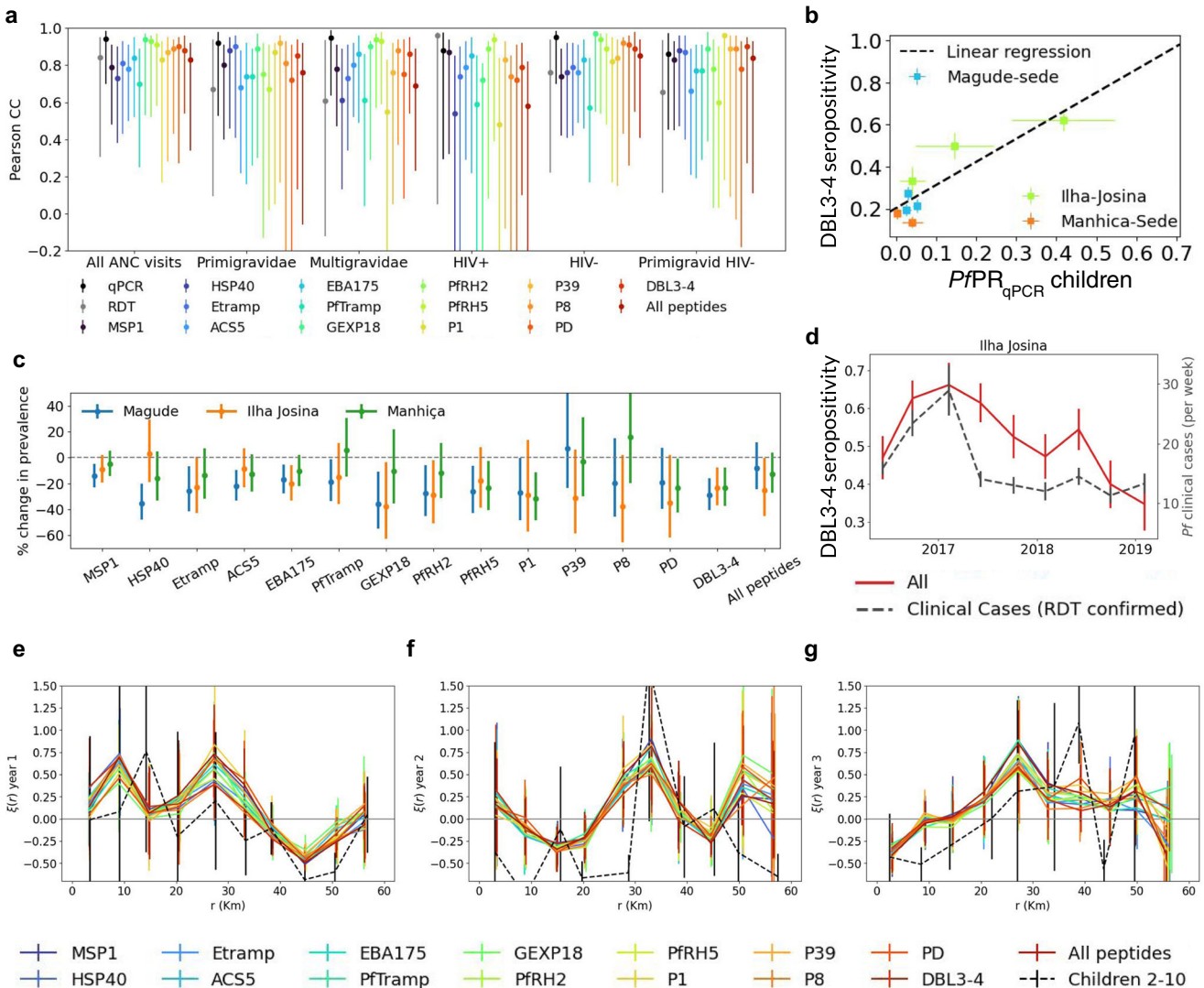

**Fig. 4 | Comparison of spatial and temporal trends in seropositive women at first antenatal care visit and *Plasmodium falciparum* infection in children.**
**a** Pearson correlation coefficients between $Pf\mathrm{PR}_{qPCR}$ in children 2–10 years old ($n = 3933$) from cross-sectional surveys and seroprevalence in pregnant women at first ANC visits ($n = 5990$). Results are shown for different gravidity and HIV status groups from pregnant women (from left to right), and each colour represents a different antigen. Grey and black data points show the Pearson correlation coefficients between children and pregnant women for $Pf\mathrm{PR}_{RDT}$ and $Pf\mathrm{PR}_{qPCR}$, respectively. **b** Scatter comparison between $Pf\mathrm{PR}_{qPCR}$ in children (x-axis) and VAR2CSA_{DBL3-4} seropositivity from all pregnant women at first ANC visit. Black dashed line shows the linear regression. **c** Declines in seroprevalence between year 1 and year 3 for each antigen expressed as percentage of change. P-values were obtained from a Z-test of proportions. **d** Comparison of the temporal trends of VAR2CSA_{DBL3-4} seroprevalence (red line) and weekly number of *Plasmodium falciparum* RDT passively detected cases from children (dashed black line) in Ilha Josina ($n = 8985$). **e–g** 2-point cross-correlation functions between seropositive cases of different antigens (represented by the different colours) ($n$ by year= 1971, 1613, 1,465) using all first ANC visits and qPCR positivity in children 2–10 years old ($n$ by year = 1900, 1565, 468) from cross-sectional surveys for the three years of study (**e–g**). Black dashed lines show the 2-point correlation function of qPCR positivity in children. Data are presented as mean values, with errors representing the 95%CI in **a** and **c**, and the SD in **b**, **d–g**.

*falciparum* infections and detected several hotspots from ANC and clinical case data. ANC data detected 60% of the clinical case hotspots. At RDT-detection levels, fewer hotspots were identified from HIV-positive and multigravidae than when including all women, probably due to lower sample size and positivity rates. Differences in the temporal distribution of hotspots were observed between ANC and clinical case data, which might result from the time lag observed, different denominator populations, inclusion of asymptomatic cases in ANC data, and variations in care-seeking behaviour. The sample size of clinical cases was limited in order to compare it with ANC data, although the larger sample size of clinical data would probably improve the precision of hotspot detection. However, passive surveillance systems usually do not record the geolocation of children routinely, precluding spatial analysis. Further studies are required to

assess the value of ANC data for identifying pockets of transmission missed by case-based surveillance and supplementing reactive strategies. Alternative geostatistical models such as kriging[22] could be explored to assess the potential of ANC data to characterise the spatial distribution of malaria prevalence in the community in case geolocation of their residence household is not available, a potentially cost-efficient approach that was previously proven successful for HIV surveillance[22]. These models could also be improved using cokriging[22] by integrating factors on climate, land vegetation and usage, spatial population distributions or even including estimations of malaria suitability surface to refine the estimates of the spatial patterns of malaria burden in the community.

Among the 14 antigens evaluated in this study, VAR2CSA_{DBL3-4}, derived from the pregnancy-specific antigen VAR2CSA[18], was found to

be the most promising marker for ANC-based sero-surveillance. VAR2CSA$_{DBL3-4}$ seroprevalence at first ANC visit showed high correlation with qPCR rates in children, across all gravidity and HIV-status groups. Temporal trends in VAR2CSA$_{DBL3-4}$ seroprevalence also showed the highest correlation with temporal trends in clinical cases. Furthermore, VAR2CSA$_{DBL3-4}$ was the only serological marker that mirrored declines of qPCR parasite rates in all three areas. VAR2CSA$_{DBL3-4}$ was able to detect declines missed by RDT, demonstrating the power of this serological marker in settings with few RDT-detectable cases. Along with several other antibodies, VAR2CSA$_{DBL3-4}$ also showed great potential to detect spatial patterns in transmission. In year 2, more sero-clusters were identified than qPCR hotspots. In particular, sero-clusters were found to persist in a low-transmission area after elimination interventions had been deployed and very few *P. falciparum* cases were detected. This shows the ability of serological markers to capture recent transmission dynamics, which would be especially useful to demonstrate continuous absence of transmission following elimination. A rapid serological test for antibodies against VAR2CSA$_{DBL3-4}$ at ANC could represent a low-cost surveillance tool with improved ability to detect trends missed by RDT, or to detect recent cases in elimination settings. Importantly, even if VAR2CSA-based vaccines under development are widely administered in the near future, VAR2CSA$_{DBL3-4}$ can still be used for sero-surveillance as it is not a vaccine target[23].

This study has several limitations. First, data collection might have been affected by RDT stock outs, changes in reporting practices[24] and human errors[25], and geolocalisation of clinical cases was only collected in Manhiça district. Second, sample sizes differ substantially between clinical, ANC and cross-sectional data, limiting the power of comparisons, especially for spatial analysis. Third, women not attending ANC tend to be older, live in rural settings, and be of lower socio-economic status than ANC-attending women, which are all risk factors for malaria[3]. However, this selection bias is expected to be limited in sub-Saharan Africa due to generally high ANC attendance[6]. Also, other factors not assessed in this study might affect the relationship between malaria in children and in pregnant women, such as other co-infections, age, usage of insecticide treated nets or gestational age of the pregnant women, and should be evaluated in future studies. Finally, single-time-point measurements may limit the ability to infer true infection and serological status due to the complex dynamics of parasites and antibody responses in the infected host[13,18]. Conducting similar studies in different epidemiological settings and varying ANC-attendance levels will be needed to confirm the generalisability of this surveillance approach.

In conclusion, malaria testing of pregnant women at their first ANC visit can provide estimates of temporal and spatial trends in malaria burden that reflect those observed in children. However, the time lag of 2–3 months relative to clinical cases, together with gravidity and diagnostic test sensitivity in high-transmission settings, need to be considered when interpreting ANC data. ANC data can also be used to detect malaria hotspots with the EpiFRIenDs algorithm, reflecting hotspots detected with passive surveillance data, potentially providing a cost-efficient approach to tailor interventions to areas most in need. EpiFRIenDs showed its superiority compared to scan statistics using SaTScan[26–28] in detecting irregular hotspots, which better reflect the spatial distribution of human populations, constituting a new tool to link foci detection with appropriate targeted approaches. Finally, antibodies against the pregnancy-specific parasite antigen VAR2CSA$_{DBL3-4}$ could serve as a more resilient marker of spatio-temporal malaria trends measured at ANC. Taken together with other potential benefits of a continuous ANC-based surveillance approach, including more precise denominator populations, and the ability to capture asymptomatic infections, surveilling pregnant women at first ANC visit has great potential to complement existing surveillance systems in Africa.

## Methods

### Study area and population

The study was conducted between November 2016 and November 2019 in Manhiça and Magude districts in Maputo Province, southern Mozambique. Malaria transmission is low in Manhiça district[29], with some moderate-to-high transmission areas, such as Ilha Josina[30]. Magude district is a low-transmission area resulting from elimination interventions since 2015[25]. Data was obtained from 8,745 pregnant women (Supplementary Fig. 4), residing in the study area, who attended their first ANC visit at Manhiça District Hospital, Ilha Josina Health Centre, or Magude Health Centre[31]. Weekly numbers of RDT-positive clinical malaria cases among children <5 years old attending the three health facilities ($n = 15,467$) were obtained from the District Health Information System 2 (DHIS2). In Manhiça district, 37,131 RDT and microscopy results from children <5 years attending health facilities were available from OPMSS. Data from 3,933 children aged 2–10 years was collected in age-stratified cross-sectional surveys conducted every May from 2015 to 2019 in the catchment areas corresponding to the three ANC health facilities of this study. Gender was not considered to define this population, providing an even representation with 50.3% of female and 49.2% of men (with 0.5% of unavailable information). Geo-localisation of the residence of pregnant women and children was obtained from a local health and demographic surveillance system using their permanent or family identification number[24,32], from their household identification number or by registering the geo-localisation of the households (Supplementary Methods).

### Recruitment and data collection

Pregnant women were only included in the study if they gave consent to participate in the study and if they met the inclusion criteria: attending a first routine ANC visit or a visit for delivery, and being from the study area with a permanent identification number issued by the demographic surveillance system or residing in the area. All the pregnant women at first ANC visit who agreed to participate were requested to donate finger prick blood onto filter paper for detection of parasite DNA and antibodies by qPCR and quantitative suspension array based on Luminex technology, respectively, together with routine tests. A standardised electronic questionnaire using REDCap was filled in with information including the visit date, age, gravidity, area of residence and recent movements. HIV status of pregnant women was determined from the maternal health card, or if not available, with an HIV serological rapid test[31]. From the 8,745 first ANC visits, 6,471 of them (74%) were analysed by qPCR (Supplementary Fig. 4), based on a random selection required to estimate annual *P. falciparum* positivity rates between 20% and 5% in each of the 3 sites, with a margin of error lower than or equal to the expected positivity rate[31]. No differences were observed between the characteristics of study participants whose samples were analysed by qPCR and those whose samples were not analysed by qPCR[31].

### Parasitological and immunological determinations

Finger prick blood drops were collected onto Whatman 903 filter paper (DBS) from pregnant women and from children in the cross-sectional surveys. Children were also tested by RDT (HRP2-based SD Bioline Ag *Pf*, Standard Diagnostics, South Korea). *P. falciparum* infection was detected and quantified in duplicate from DBS with a qPCR targeting the 18 S rRNA gene (using primers "18s-QF": 5′ GTA ATT GGA ATG ATA GGA ATT TAC AAG GT 3′ and "18s-QR": 5′ TCA ACT ACG AAC GTT TTA ACT GCA AC 3′) on an ABI PRISM 7500 HT Real-Time System (Applied Biosystems)[33]. Immunoglobulin Gs (IgG) were detected and quantified in DBS ($n = 6038$) using a multiplexed bead array, the xMAP© technology and the Luminex 100/200 System (Luminex Corp., Austin TX)[31]. In brief, magnetic beads were coupled to our panel (Supplementary Table 12) including VAR2CSA antigens (Duffy binding-like recombinant domains DBL3-4, peptide P1 targeting the NTS

region, peptides P8 and PD targeting ID1, and P39 targeting DBL5ε), general malaria antigens (19-kDa fragment of the merozoite surface protein-1 [MSP$_{19}$], region II/F2 of erythrocyte-binding antigen-175 [EBA175], full-length *P. falciparum* reticulocyte binding-like homologue protein 2 and 5 [RH2 and RH5]), thrombospondin-related apical merozoite protein [PfTRAMP], and biomarkers of recent *P. falciparum* exposure (gametocyte exported protein 18 [GEXP18], acyl-CoA synthetase 5 [ACS5] ag3, early transcribed membrane protein 5 [ETRAMP5] ag1, and heat shock protein 40 [HSP40] ag1). Information about antigens, procedures for reconstitution of DBS and quality control, bead-based immunoassay, and data normalisation are described in Supplementary Methods.

### Data analysis

Study year 1 to 3 were defined from November to October in 2016–2017, 2017–2018 and 2018–2019, respectively. Infections with densities above 100 parasites/μL were defined as RDT-detectable[34]. Primigravity was defined as a first pregnancy, and multigravidity as having had one or more previous pregnancies. The threshold of seropositivity against *P. falciparum* antigens was defined as the geometric mean plus 2 standard deviations of the first component from two-component normal mixture distributions of mean fluorescent intensity values (R package *mixtools*).

The relationship between $Pf$PR$_{RDT}$ or $Pf$PR$_{qPCR}$ detection limits in pregnant women and children was analysed using linear regressions, PCC, and $\chi^2$ statistics (Supplementary Methods). Similarly, anti-*P. falciparum* seroprevalence in pregnant women was compared with $Pf$PR$_{RDT}$ and $Pf$PR$_{qPCR}$ in children. Consistency and correlation between temporal variations in both populations were quantified using $\chi^2$ statistics and PCC, and time lags between the data sources were defined by maximising PCC (Supplementary Methods). 2PCFs were used for clustering analysis, which describe the excess of pairs of *P. falciparum* positive samples with respect to random infections as a function of the geographical distance $r$ between them:

$$\xi(r) = \frac{P_1P_2(r)}{B_1B_2(r)} - 1, \qquad (1)$$

where $\xi(r)$ is the 2PCF at distance $r$, $P_1P_2(r)$ is the number of pairs of positive cases between populations 1 and 2 separated a distance $r$ between them, and $B_1B_2(r)$ is the number of background pairs (positive or negative) between both populations at distance $r$. $P_1P_2(r)$ and $B_1B_2(r)$ were normalised by $nP_1nP_2$ and $nB_1nB_2$ respectively, where $nP_{1,2}$ and $nB_{1,2}$ are the number of positive samples and the total number of samples respectively for population 1,2. The 2PCF measurements were done using 10 bins in distance in a range from 0 to 60 km. The choosing of these bins was based on the spatial range of pairwise distances and finding the balance between spatial granularity and the statistical power (sample size) of the measurements. The agreement between different 2PCFs was quantified with $\chi^2$ statistics.

A novel malaria hotspot detector, EpiFRIenDs, was developed to detect areas with higher levels of *P. falciparum* infections (hotspots) and seropositivity against *P. falciparum* antigens (seroclusters) than statistically expected in a stable period of time[2]. EpiFRIenDs was designed to detect structures of arbitrary shapes and sizes that account for the background population distribution, a difference with the most commonly used scan statistics based on the SaTScan software[26-28] that detects structures of pre-defined shapes. EpiFRIenDs detects hotspots and seroclusters by linking positive cases when they are closer than a given pre-defined distance and indirectly linking them to all the positive cases that are close to their connections. The negative cases are then included in the hotspots from their close positive cases. The EpiFRIenDs software is publicly available in Python[35] and R[36] and it is described in detail in the section EpiFRIenDs of Supplementary Methods. The hotspot detection using EpiFRIenDs and SaTScan (scan

statistics) was first compared in simulated data with a prevalence of 20% reproducing three different scenarios: first, a random spatial distribution of positive and negative cases; second, four circular clusters of positive cases on top of a background random distribution of negative cases; and third, a sinusoidal distribution of positive cases on top of a background random distribution of negative cases. EpiFRIenDs was then applied to identify hotspots and serological clusters from ANC, OPMSS and antibody data (Supplementary Methods). Hotspots were compared between ANC and clinical data from the Manhiça District Hospital and the Ilha Josina Health Centre, for which geolocation information was available. Since EpiFRIenDs is a density-based clustering algorithm, clinical data was randomly sub-sampled from the 37,131 visits recorded in Manhiça district to obtain the same sample size of positive cases than for ANC data. In order to obtain a statistical significance of the differences between the hotspots detected from ANC and from clinical cases, EpiFRIenDs was run on five hundred different random sub-samples of the clinical data. Linking distance (1 km) and temporal windows (one month or one year) were defined based on sample density. Hotspot size thresholds (three or five positive cases per hotspot) were used to avoid false detections, and the sizes were estimated from 500 realisations, where in each realisation all data was used with their corresponding geographical location, but assigning them infections with a probability based on their overall prevalence (Supplementary Methods). A hotspot from ANC data was considered to be spatially matched by a hotspot from clinical data (or vice versa) if at least one member of an ANC hotspot was found closer than 2 km to a member of a hotspot from clinical data, and was considered to be spatio-temporally matched if in addition the temporal separation between hotspots was shorter than 60 days.

All analyses were stratified by gravidity and HIV status, the main factors shown to affect $Pf$PR$_{qPCR}$ in our study population (Supplementary Table 13), with primigravid HIV-negative women considered separately to take potential correlations between gravidity and HIV into account. Statistical significance was set at $p < 0.05$. $\chi^2$ values were interpreted as good consistency from thresholds corresponding to $p < 0.05$ in a null-hypothesis statistical test (Supplementary Methods). Error bars and 95% confidence intervals were obtained from bootstrap resampling with replacement. The analysis was conducted using Python 3.8.12, Jupyter Lab 3.1.14, R 4.2.1 and EpiFRIenDs 1.0.

### Reporting summary

Further information on research design is available in the Nature Portfolio Reporting Summary linked to this article.

## Data availability

The raw epidemiological, clinical and GPS data are protected and are not available due to data privacy laws. The datasets generated and/or analysed during the current study are available under restricted access for data privacy laws involving personal data. Access can be obtained for research purposes by requesting to the corresponding author by email, specifying contact details, affiliation and purpose of the request. All requests for processed data used in this study are reviewed in a three-month timeframe by Manhiça Health Research Center to verify if the request is subject to any intellectual property or confidentiality obligations. Patient-related data not included in the paper might be subject to patient confidentiality. Any data that can be shared will be released via a Data Transfer Agreement.

## Code availability

All the code used in the analysis is open source under a GNU General Public License. The main open source repository of this paper can be accessed here: https://github.com/arnaupujol/anc_surveillance_tools[37]. The repository contains software requirements and installation instructions, with references to other public repositories including EpiFRIenDs[35,36].

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

## Acknowledgements

We are grateful to the women, children and their families who participated in the study, the clinical teams at the health centres, the field and lab teams at CISM (Henrique Cossa, Judice Miguel, Manuel Muamede, and Helga Guambe) and ISGlobal (Laura Puyol, Diana Barrios, Alfons Jiménez, Marta Vidal, Rebecca Santano, Martina Fort Suñé, Marta Menendez García, Gemma Porras, Haily Chen, Elena Buetas, and Ianthe de Jong). We would like to express our gratitude to the communities of Magude and Manhiça districts and the district authorities for their support of the project. We also thank Patrick Walker and Joseph Hicks for the fruitful discussion on this project, and Gillian Stresman, Maria Tusell, Arantxa Roca-Feltrer, and Dídac Macià for useful discussions on the design of EpiFRIenDs. This work was supported by the National Institute of Health (1R01AI123050, E.M./A.M.), the Bill and Melinda Gates Foundation (INV-019032, A.M.), the Departament d'Universitats i Recerca de la Generalitat de Catalunya (AGAUR; grant 2021 SGR 01517, A.M.),

Ministerio de Ciencia e Innovación (PID2020-118328RB-I00, A.M.), the European Union's Horizon 2020 research and innovation programme under the Marie Skłodowska-Curie grant agreement No 890477 (A.P.) and from the "la Caixa" Foundation (ID 100010434, fellowship code LCF/BQ/DI20/11780016, N.B.). ISGlobal is a member of the CERCA Programme, Generalitat de Catalunya (http://cerca.cat/en/suma/). CISM is supported by the Government of Mozambique and the Spanish Agency for International Development (AECID). We acknowledge support from the grant CEX2018-000806-S funded by MCIN/AEI/10.13039/501100011033, and support from the Generalitat de Catalunya through the CERCA Program. This research is part of the ISGlobal's Program on the Molecular Mechanisms of Malaria which is partially supported by the Fundación Ramón Areces. The funders had no role in study design, data collection and analysis, decision to publish, or preparation of the manuscript.

## Author contributions

A.M. and E.M. designed the study. S.M. and H.Mbeve recruited study participants at health facilities. B.C., C.M. and N. N. managed the data and samples. B.Galatas (2015–2017), C.G., H.Munguambe (2015–2019), L.N. (2015–2018), W.S. (2015–2019), J.Montaña (2018–2019), F.S. and P.A. coordinated the cross-sectional studies recruiting community participants. A.C., H.M.-S., B.Galatas, L.Q. and J.Montaña managed the data from cross-sectional studies. N.B., G.M., P.C. and A.E., analysed the samples. K.T., B.Gamain, C.C., C.D. and V.C. produced recombinant antigens for serological determinations. A.P. and N.B. analysed and interpreted the data. A.M. and G.M. contributed to the interpretation of data. A.P. and N.B. drafted the manuscript. All authors critically revised the manuscript, had full access to the data used in the study, and accepted the responsibility to submit it for publication.

## Competing interests

The authors declare no competing interests.

## Ethical approval

All study protocols were approved by the institutional ethics committees at CISM and Barcelona Hospital Clínic, and the Mozambican Ministry of Health National Bioethics Committee.

## Informed consent

Only participants giving consent were included in this study. All data and biological samples were collected only if research participants (or representatives in the case of minors) gave full written informed consent.
