## [Peer Review File · Nature Communications]

Detecting temporal and spatial malaria patterns from first antenatal care visitsREVIEWER COMMENTS

Reviewer #1 (Remarks to the Author):

This is an important and original study, validating the use of malaria test results at first ANC visits as an indicator of malaria control in the area. The authors have attempted to look from different angles, using different test (RDT, PCR, serology), different comparison groups (sick children visits, or cross-sectional child data), and different methodologies (over time, by time and space), and among different subgroups of pregnant women (HIV-status, by gravidity). This study is important, because it informs if and how pregnant women can be used to assess status of malaria control in a region in a timely manner. The manuscript is dense with information, which makes it not straightforward to understand. I do recommend publication because this will further the development of indicators of malaria control in areas with low transmission. I only have few comments.

Methods

Just to make sure I understand it properly: PCR data was obtained from pregnant women, and PCR and RDT data was obtained from children in the cross-sectional surveys. To be able to compare RDT data from children with pregnant women, the pregnant women PCR results were recalibrated using densities above 100 parasites per microliter.

Results

The time lag is a bit confusing: perhaps a short explanation below the figure or in the text may help. E.g., with table 1: "A time lag of 89 days indicates that the results of the clinical case in children had the best correlation with the results of pregnant women taken 89 days before the clinical cases in children." As it is, it is not clear if antenatal women can predict the clinical cases, or if clinical cases will predict prevalence in pregnancy later.

Figure S1: 8745 First ANC visits, 6471 qPCR (74%). What was the reason for this discrepancy and could this have introduced bias?

Discussion

"However, temporal trends in ANC-based parasite rates, both detected by qPCR and RDT, reflected trends in clinical cases observed 2-3 months earlier, similar to the 3-month time lag observed in the Democratic Republic of Congo." So does this mean that ANC prevalence cannot be used to predict an increase in clinical cases? Or could clinical cases be used to shore up malaria-in-pregnancy control? Might the time lag depend on gestational age at first visit? Or be affected by ITN use?

Reviewer #2 (Remarks to the Author):

Review of report for the manuscript "Detecting temporal and spatial malaria patterns from first antenatal care visits"

General comment

The work is intensively done, providing relevant and useful information to improve malaria surveillance in endemic countries.

I battled with if the 'experimental' exercise assessing the performance of the new tool to detect hotspots [EpiFRiEnDs] against SaTScan should be included as part of this manuscript. Understandably of the importance of innovative tools for malaria surveillance in MECs, it made the paper to be too heavy, in my opinion, and somehow deviating the reader from the core contribution of this work, i.e., assessing the usefulness of ANC malaria surveillance data. However, that doesn't lessen the incredible value of the epidemiological findings - leave it to the team.

Specific comments

Methods

- HIV status for the pregnant women - In lines 103 and 137 a reference is used on how the test

results were obtained; That being all good, I would recommend expansion of this section summarizing what was done in the prospective observational study for clarity. As it is presented now it creates a methodological gap to understand the process.

- Hotspot detection comparing ANC and clinical data - Authors described that 500 realizations were done to obtain subsamples for the clinical data, however, all these being randomly selected create skeptics on how already existing structural patterns in the clinical data were captured in the subsamples generated considering the underlying processes generating these data as described by authors (treatment seeking, symptomatic cases, testing rates, etc). What does 'Keeping their geographical distribution' explained in the Supp material means? Could this be expanded in the main paper?

Results

Figure 1 (applies also to Figure S2) Scatter comparison do not indicate the key for the three points for each site (which are supposed to represent different values for each year?) or are these sequentially ordered?

Minor error in Figure S1. - under Luminex for I.Josina (Y3) 1959 may be should be 195?

Line 267: Seroprevalence of antibodies against *P. falciparum* antigens

Lines 269-271 - Should this be presented in ordered manner of time assuming either the seroprevalence would be increasing or decreasing and probably evolve - saying "... ranged ... during Year 3 to ... during first year" sounds a bit confusing.

Discussion and Conclusion

Lines 379 - 382 I would shifted the statement about "Bias" to discussion section. Doesn't fit in the conclusion very well.

Reviewer #3 (Remarks to the Author):

This study attempts to show the utility of ANC data for the detection of space-time clusters of malaria prevalence mirrored against health facility and underlying community based survey data as well as applies a novel clustering technique to detect irregular shaped clusters for malaria. This work makes a strong contribution to this area of work and addresses some of the methodological and other gaps identified in previous studies. The findings suggest that ANC data may provide utility in low transmission settings and in particular among primigravida women. These data also appear to accurately approximate underlying temporal trends for malaria burden reductions in this study area.

Some comments below which I hope will be of use:

For the spatial analysis of ANC visits, just confirming the reported residence location of the women was used? Assuming this approach was applied in other locations using ANC data without residence location, how could the ANC facility location be used to infer underlying community based malaria prevalence distribution? This could be done using a kriging/geostatistical model e.g. Cuadros DF, Sartorius B, Hall C, Akullian A, Bärnighausen T, Tanser F. Capturing the spatial variability of HIV epidemics in South Africa and Tanzania using routine healthcare facility data. *International Journal of Health Geographic's*. 2018 Dec;17(1):1-9.

Following on the above point, this would be more a discussion point but could a malaria suitability surface be further incorporated to refine estimates of underlying community malaria trends from facility/clinic data?

Spatial trends/hotspots - I think it would be useful to include some additional metrics showing how much of the detected hotspots overlap between ANC and child survey clusters in terms of area (space) and area-time.

We are very grateful to the reviewers for the time spent in revising the document and providing constructive comments. We have addressed them all and hope that the quality of the manuscript increases in this revised version.

Reviewer #1:

This is an important and original study, validating the use of malaria test results at first ANC visits as an indicator of malaria control in the area. The authors have attempted to look from different angles, using different test (RDT, PCR, serology), different comparison groups (sick children visits, or cross-sectional child data), and different methodologies (over time, by time and space), and among different subgroups of pregnant women (HIV-status, by gravidity). This study is important, because it informs if and how pregnant women can be used to assess status of malaria control in a region in a timely manner. The manuscript is dense with information, which makes it not straightforward to understand. I do recommend publication because this will further the development of indicators of malaria control in areas with low transmission. I only have few comments.

Methods

Comment: Just to make sure I understand it properly: PCR data was obtained from pregnant women, and PCR and RDT data was obtained from children in the cross-sectional surveys. To be able to compare RDT data from children with pregnant women, the pregnant women PCR results were recalibrated using densities above 100 parasites per microliter.

Answer: Exactly, this is correct. *P. falciparum* infection in pregnant women was tested by qPCR, and RDT detectability calculated from parasite densities estimated in the molecular assay. Children were tested both by qPCR and RDT.

Results

Comment: The time lag is a bit confusing: perhaps a short explanation below the figure or in the text may help. E.g., with table 1: “A time lag of 89 days indicates that the results of the clinical case in children had the best correlation with the results of pregnant women taken 89 days before the clinical cases in children.” As it is, it is not clear if antenatal women can predict the clinical cases, or if clinical cases will predict prevalence in pregnancy later.

Answer: We have added the following text to clarify the concept. In Table 1, we added “A time lag of 89 days indicates that the results of the clinical cases in children had the best correlation with the results of pregnant women taken 89 days after the clinical cases in children”. In the caption of Figure 2, we added a reference to Table 1. In the results section, we added the comment “meaning that the best correlation was obtained between clinical cases and ANC data collected ~90 days after the clinical cases”.

Comment: Figure S1: 8745 First ANC visits, 6471 qPCR (74%). What was the reason for this discrepancy and could this have introduced bias?

Answer: We estimated that the molecular detection of *P. falciparum* in all dried blood spots collected from Ilha Josina (n=250/year) and a random selection of approximately 2700 samples per year in both Magude and Manhiça would allow for the estimation of the 95% confidence interval (95%CI) of annual *P. falciparum* positivity rates between 20 and 5% in each of the 3 sites, with a margin of error lower than or equal to the expected positivity rate (as detailed in Matambisso, G. et al. Gravidity and malaria trends interact to modify *P. falciparum* densities and detectability in pregnancy: a 3-year prospective multi-site

observational study. *BMC Med* 20, 396, 2022). The selection of the samples was done randomly. We checked that no differences were observed between the characteristics of study participants whose samples were analysed by qPCR and those whose samples were not analysed by qPCR (Matambisso, G. *et al.*, Additional File 3: Table S1). This information is now added in the methods section.

Discussion

Comment: “However, temporal trends in ANC-based parasite rates, both detected by qPCR and RDT, reflected trends in clinical cases observed 2-3 months earlier, similar to the 3-month time lag observed in the Democratic Republic of Congo.” So does this mean that ANC prevalence cannot be used to predict an increase in clinical cases? Or could clinical cases be used to shore up malaria-in-pregnancy control? Might the time lag depend on gestational age at first visit? Or be affected by ITN use?

Answer: The time lag means that the changes observed from clinical cases are observed later in time from ANC data. On the other hand, the results imply that clinical cases can be used to predict changes in ANC data. We added the following sentences in the discussion section: “Although the time lag means that ANC data cannot be used to predict changes in clinical cases, it would still be useful to benchmark passive surveillance estimates and population-based cross-sectional surveys, therefore improving population denominators and informing about the burden of asymptomatic infections. On the other hand, the results imply that clinical cases can be used to predict changes in malaria burden among pregnant women at first ANC visit.” Regarding the potential dependence of gestational age and ITN usage on the time lag, we did not explore these factors explicitly. We followed an analytical plan based on the key factors (HIV status, test detectability and gravidity) identified in our previous study (Matambisso, G. *et al.* Gravidity and malaria trends interact to modify *P. falciparum* densities and detectability in pregnancy: a 3-year prospective multi-site observational study. *BMC Med* 20, 396 (2022)). Moreover, ITN data was not collected during the first months of the study, and therefore data is incomplete. However, we agree with the reviewer about the possible role of these factors, so we have now included this sentence in the discussion: “Also, other factors not assessed in this study might affect the relationship between malaria in children and in pregnant women, such as other co-infections, age, usage of insecticide treated nets or gestational age of the pregnant women, and should be evaluated in future studies. “

Reviewer #2:

General comment

The work is intensively done, providing relevant and useful information to improve malaria surveillance in endemic countries.

Comment: I battled with if the 'experimental' exercise assessing the performance of the new tool to detect hotspots [EpiFRlenDs] against SaTScan should be included as part of this manuscript. Understandably of the importance of innovative tools for malaria surveillance in MECs, it made the paper to be too heavy, in my opinion, and somehow deviating the reader from the core contribution of this work, i.e., assessing the usefulness of ANC malaria surveillance data. However, that doesn't lessen the incredible value of the epidemiological findings - leave it to the team.

Answer: Thank you for valuing our findings. This is in fact something that we discussed a lot, whether to include the comparison and exploration of the two hotspot detectors (EpiFRlenDs and SaTScan) in the manuscript. On one hand, it is true that this adds substantial content to the manuscript, making it heavier with a contribution that is not directly connected to the assessment of ANC malaria surveillance data. However, since this manuscript is the first one using the new method EpiFRlenDs, presenting the results without a deeper analysis of the method would question the validity of the data presented. For this reason, we decided to describe the EpiFRlenDs methodology as a supplementary material. We would prefer to keep as it is, in order to convince the readers about the robustness of the spatial analysis presented in the manuscript.

Specific comments

Methods

Comment: HIV status for the pregnant women - In lines 103 and 137 a reference is used on how the test results were obtained; That being all good, I would recommend expansion of this section summarizing what was done in the prospective observational study for clarity. As it is presented now it creates a methodological gap to understand the process.

Answer: We have expanded the details regarding the recruitment and data collection of pregnant women with an extra section called "Recruitment and data collection". Part of the text has been moved here from the supplementary material.

Comment: Hotspot detection comparing ANC and clinical data - Authors described that 500 realizations were done to obtain subsamples for the clinical data, however, all these being randomly selected create skeptics on how already existing structural patterns in the clinical data were captured in the subsamples generated considering the underlying processes generating these data as described by authors (treatment seeking, symptomatic cases, testing rates, etc). What does 'Keeping their geographical distribution' explained in the Supp material means? Could this be expanded in the main paper?

Answer: We applied 500 random realisations in different scenarios. First, in order to apply a hotspot size cut to avoid false detection. Second, to create different sub-samples of clinical cases in order to assess the statistical significance of the differences between the detected hotspots from ANC and clinical cases. In the first case, by "keeping their geographical distribution" we mean that each realisation uses the exact same data, with their preserved geographical positions, and we only change the test result by assigning a probability of being positive from their overall positivity rates. In this way, the spatial distribution of the population is preserved, and the differences in their distribution of positive cases allow us to identify false detections under the assumption that the test result does not depend on the geographical location (i.e. there is no clustering of infections). In the second case, each sub-sampling of clinical cases is obtained by selecting a random sub-selection from the whole dataset. In both cases, the structural patterns of the data are statistically kept representative over the 500 realisations and no biases are expected with respect to the original data. To clarify these aspects, we have added the following sentences in the Methods section:

- "In order to obtain a statistical significance of the differences between the hotspots detected from ANC and from clinical cases, EpiFRlenDs was run on five hundred different random sub-samples of the clinical data."
- "Hotspot size thresholds (three or five positive cases per hotspot) were used to avoid false detections, and the sizes were estimated from 500 realisations, where in each

realisation all data was used with their corresponding geographical location, but assigning them infections with a probability based on their overall prevalence.”

We have also added the following sentences in the Supplementary Methods:

- “The size threshold was applied to include only keep hotspots with at least three positive cases. The size threshold was applied in order to avoid false detections, and was defined by running 500 realisations of the datasets. In each realisation, all data was kept, with their geographical distribution but assigning infections from a probability defined by their overall prevalence. This allowed to study the detection of hotspots in the scenario that infections are not correlated with geographical location (i. e. there is no clustering of infections).”
- “In each of the realisations, a different random selection of 18% of the clinical data was used, so that the sample size was kept without biasing the properties of the sub-samples.”

Results

Comments: Figure 1 (applies also to Figure S2) Scatter comparison do not indicate the key for the three points for each site (which are supposed to represent different values for each year?) or are these sequentially ordered?

Answer: Yes, each of the three points of each colour represent a measurement for each year, corresponding to the three years for which the cross-sectional studies were conducted. We clarified this in the caption of the figures.

Comment: Minor error in Figure S1. - under Luminex for I.Josina (Y3) 1959 may be should be 195?

Answer: Yes, this is an error, thank you very much for noticing. We fixed it in the new version.

Comment: Line 267: Seroprevalence of antibodies against *P. falciparum* antigens
Lines 269-271 - Should this be presented in ordered manner of time assuming either the seroprevalence would be increasing or decreasing and probably evolve - saying "... ranged ... during Year 3 to ... during first year" sounds a bit confusing.

Answer: Thank you for noticing this confusing sentence, we agree that it is not clear enough. We rephrased it, we hope it is now clearer: “The lowest seroprevalence was found for VAR2CSAP8 in Magude during the third year (3.9%, 95%CI 3.0-5.1), and the highest seroprevalence was found for MSP1 in Ilha Josina during the first year (76.8%, 95%CI 72.3-80.8)”

Discussion and Conclusion

Comment: Lines 379 - 382 I would shifted the statement about "Bias" to discussion section. Doesn't fit in the conclusion very well.

Answer: We have moved the sentence to the discussion of results, we hope it is better suited now.

Reviewer #3:

This study attempts to show the utility of ANC data for the detection of space-time clusters of malaria prevalence mirrored against health facility and underlying community based survey

data as well as applies a novel clustering technique to detect irregular shaped clusters for malaria. This work makes a strong contribution to this area of work and addresses some of the methodological and other gaps identified in previous studies. The findings suggest that ANC data may provide utility in low transmission settings and in particular among primigravida women. These data also appear to accurately approximate underlying temporal trends for malaria burden reductions in this study area.

Some comments below which I hope will be of use:

Comment: For the spatial analysis of ANC visits, just confirming the reported residence location of the women was used? Assuming this approach was applied in other locations using ANC data without residence location, how could the ANC facility location be used to infer underlying community based malaria prevalence distribution? This could be done using a kriging/geostatistical model e.g. Cuadros DF, Sartorius B, Hall C, Akullian A, Bärnighausen T, Tanser F. Capturing the spatial variability of HIV epidemics in South Africa and Tanzania using routine healthcare facility data. *International Journal of Health Geographic's*. 2018 Dec;17(1):1-9.

Answer: Yes, the geographical location of the participants corresponds to the residence location, for both women and children. We state it explicitly now in the section “Study area and population”. The observation regarding potential approaches in case that no residence household location is very much appreciated, and we added some discussion about this possibility in the discussion section, citing the reference suggested by the reviewer.

Comment: Following on the above point, this would be more a discussion point but could a malaria suitability surface be further incorporated to refine estimates of underlying community malaria trends from facility/clinic data?

Answer: This is a very interesting observation. Models of malaria suitability surface could be integrated as a factor in cokriging analyses to refine the predictions of the spatial distribution of malaria burden in the community from sparse data, such as ANC data without residence location available. We added some sentences to discuss this idea in the discussion section, and we thank the reviewer for the interesting insights.

Comment: Spatial trends/hotspots - I think it would be useful to include some additional metrics showing how much of the detected hotspots overlap between ANC and child survey clusters in terms of area (space) and area-time.

Answer: We thank the referee for this constructive recommendation. We added the new Supplementary table S7 indicating the fraction of hotspots spatially and spatio-temporally matched between the two data sources, explained the matching criteria in the methods section and described the results in the results section.

REVIEWERS' COMMENTS

Reviewer #1 (Remarks to the Author):

The authors have responded to my comments satisfactorily. I recommend publication.

Reviewer #2 (Remarks to the Author):

Thank you for giving me the opportunity to revise this important manuscript.

With the increasing need of using surveillance data as a tool for monitoring malaria, guiding optimal use of resources and effectively applying interventions in time and space, this work provides a significant utility to the field. I specifically commend the innovation of the new hotspot detector EpiFRiendS.

I acknowledge all the work done by the authors in responding to the comments raised. My comments were addressed adequately.

Extra revisions and additions noted in e.g. Introduction, Methods, Supplementary materials sections of the manuscripts added more clarity to the paper, awesome!

A small observation to Authors: I don't find the PDF files for Supplementary Movies 1, 2 and 3 i.e., ones accompanied the GIF to be easy to follow/understand, hence not sure if are necessary, unless these are needed by the Journal.